# Fostering Cyber-Physical Social Systems through an Ontological Approach to Personality Classification Based on Social Media Posts

**DOI:** 10.3390/s21196611

**Published:** 2021-10-04

**Authors:** Alexandra Cernian, Nicoleta Vasile, Ioan Stefan Sacala

**Affiliations:** Faculty of Automatic Control and Computers, University Politehnica of Bucharest, 060042 Bucharest, Romania; nicoleta.vasile@stud.acs.upb.ro (N.V.); ioan.sacala@upb.ro (I.S.S.)

**Keywords:** cyber-physical social system, ontology, DISC personality model, personality classification, social media posts, statement analysis, Romanian language

## Abstract

The exponential increase in social networks has led to emergent convergence of cyber-physical systems (CPS) and social computing, accelerating the creation of smart communities and smart organizations and enabling the concept of cyber-physical social systems. Social media platforms have made a significant contribution to what we call human behavior modeling. This paper presents a novel approach to developing a users’ segmentation tool for the Romanian language, based on the four DISC personality types, based on social media statement analysis. We propose and design the ontological modeling approach of the specific vocabulary for each personality and its mapping with text from posts on social networks. This research proposal adds significant value both in terms of scientific and technological contributions (by developing semantic technologies and tools), as well as in terms of business, social and economic impact (by supporting the investigation of smart communities in the context of cyber-physical social systems). For the validation of the model developed we used a dataset of almost 2000 posts retrieved from 10 social medial accounts (Facebook and Twitter) and we have obtained an accuracy of over 90% in identifying the personality profile of the users.

## 1. Introduction

### 1.1. The Problem in the Current Context

The exponential increase in social networks has led to emergent convergence of cyber-physical systems (CPS) and social computing, accelerating the creation of smart communities and smart organizations and enabling the concept of cyber-physical social systems [1]. This will set new standards for the interactions and cooperation between people, objects and organizations that interact and cooperate with each other, while also leveraging innovative applications and services that will improve quality of life and work. In this context, knowing people and understanding their behavior will become increasingly important in incorporating cyber space, physical space and social space. While cyber space and physical space have been more in the focus of researchers for a long time, there is still a lack in investigating the social space from a data-driven perspective.

Over the years, the progress and evolution of technology have shaped human behavior that has adapted to the change of the digital age. Social media platforms have become a substantial element of the present and have made a significant contribution to what we call human behavior modeling, as our lives focus on sharing opinions, thoughts and information with those around us through these platforms. These activities are stored as digital traces and can provide an essential perspective on the lives of individuals, including details that are supposed to be private and personal. Therefore, they can facilitate the analysis, identification of patterns and outlining conclusions about who we are, what we are, what and how we do. The textual content of our online presence reveals a great deal about the personality of users [2].

Personality sums up a set of characteristics that mentally and behaviorally shape a person and distinguish him as a conscious and free individual [3]. Due to its consistency over the years and consistency in predicting results related to psychological well-being, mental health, physical health and choosing a successful career, personality is considered one of the most extensive areas of research in Psychology.

### 1.2. Our Solution and the Objectives of Our Research

The main goal of the current research is to extend the mechanism of sentiment analysis through behavioral patterns of users in social networks and to identify the behavioral profile of users by analyzing statements in social media for the Romanian language. The innovative element consists of framing users in different behavioral and personality patterns following the analysis of their statements in social networks, by using a more detailed classification of sentiment analysis and integrating semantic technologies for semantic analysis of feelings in social networks.

This paper presents a novel approach to developing a users’ segmentation tool for the Romanian language, based on the four DISC personality types [4] and leveraging social media statement analysis. We address a semantic classification of individuals based on the DISC model, whose abbreviation comes from the four personality types it encompasses: Dominant, Influential, Conscious and Stable. We propose and design the ontological modeling approach of the specific vocabulary for each personality and its mapping with the text from posts on social networks. The ontological model for DISC is based on Romanian vocabulary, which is the only one of its kind, and thus brings an innovative note to the study. The classification of the four personality types is made on two scales: orientation towards people or towards the accomplishment of tasks and own fast or moderate work rhythm [5].

Considering the use of the DISC personality model, this application aims to be appropriate to be used in employee recruitment processes, as this model offers a valuable and professional perspective on the profile of the employee, providing insights on what motivates them, what work environment they prefer or how they collaborate. Another use case we target is for marketing analytics purposes, which segments the target audience into the four DISC personality types, thus allowing marketing specialists to create customized campaigns for each category. Companies are turning to sentiment analysis [6] to learn what customers think and to boost customer-centric strategies for sales and marketing campaigns. Marketing departments rely on using predictive analytics and devising marketing campaigns based on various criteria to improve brand awareness and loyalty and to approach consumer interactions at a higher and more personal level [6], which boosts the development of smart communities around organizations.

This research proposal adds significant value both in terms of scientific and technological contributions (by developing semantic technologies and tools), as well as in terms of business, social and economic impact (by developing a tool for decision support and predictive analysis in recruitment and marketing analytics). We focus this effort on the Romanian language, where little effort has been made regarding natural language processing and therefore, recruitment and marketing specialists cannot rely on automatic tools for candidates and customer segmentation based on semantic criteria.

The topic of interest is semantics, namely building a model related to the vocabulary used by each DISC personality typology. This model will be used to analyze social media posts from Facebook or Twitter and will identify what type of personality each user has. Then, based on the result, the application can make recommendations about what the user’s behavioral patterns and preferences are, as well as recommendations of how he/she should be approached. This automated method of personality analysis would certainly reduce time and cost resources and bring a note of authenticity and objectivity.

## 2. State of the Art Analysis

### 2.1. Our Solution and the Objectives of Our Research

Personality is defined as a set of characteristics of a person, including how he acts, how he thinks and how he feels. Personality correlates with emotions, values, attitude and even talents. These attributes establish and define unique personalities for everyone, thus managing to differentiate ourselves from each other [7]. According to Stemmler, a person′s personality is closely related to the language used in writing or speaking, because this is the main form of expression and externalization [1]. Language is the most widespread and reliable tool that people have at hand to convey their thoughts and emotions.

Therefore, the analysis of the human being from a linguistic point of view is an essential and interesting topic in the field of psychology and communication. There are studies and research papers that have examined the personality of the human being using language as the main tool, and these will be described in the following.

### 2.2. Ontologies

Ontology is a set of concepts that can model the concepts in a field of knowledge [8]. The components of ontology are classes, sometimes called concepts, the properties and roles of each concept that describe the characteristics and attributes of that concept, the constraints, the instances represented by words or phrases specific to the field discussed and the relationships between instances and classes [8].

The reasons for using the construction of an ontological model will be the following:To share a common understanding of the structure of information between software developers.To allow the reuse of knowledge in a field.To separate field knowledge from operational knowledge.To explain the knowledge of a field.To analyze a field of knowledge.

The concept of ontology has three paradigms [8] regarding the construction of the model:Top-Down: starts with the definition of the most general concept and then its subsequent specializations.Bottom-Up: starts by defining the most specific class, the leaves of the hierarchy, then the subsequent grouping of these classes into more general concepts.Combination: is a combination of the two models above, in which the more obvious concepts are defined, then generalized and specialized accordingly.

For this study, the first development model was used.

The tools available for the development of ontologies are vast, as are the related languages, which have emerged as an evolution of existing languages for knowledge representation. Among the most used ontological languages I mention the following:XML—eXtensible MarkUp Language [9]—facilitates the creation of tags for applications, the meanings of which must be known by the developers in advance.RDF—Resource Description Framework [10]—developed by the W3C and is based on the idea of a triplet that must contain a description for a concept, a description of the properties of the concept and a description of the values of these properties.OWL—Ontology Web Language [11]—is built for XML and RDF standards and expands them with a richer vocabulary to describe the concepts, attributes and relationships between these notions.

Various recent studies have shown that ontology is able to model knowledge for measuring personality. There have been studies that calculate the derivation of the human personality in the field of physiognomy or research that has built an ontological model for the vast field of music to see how it relates to the field of personality [1]. These studies and research are intended to clarify how ontology could contribute to the analysis, understanding and research of human behavior and psychological research. A more detailed analysis of these studies and a comparison with the current study will be made in the following section.

### 2.3. Related Work

In this section we will discuss existing research in the field of personality study. We have found that there are many studies aimed at linking personality types and activity on social networks. For example, people with a high score for neurosis use social media services such as Facebook, Twitter and blogs more often [12,13].

In 2000, Amichai-Hamburger and Ben-Artzi [14] analyzed Internet use patterns for 45 men and 27 women with extrovert and neurotic traits. According to this study, for men, extraversion obtained a positive score in terms of relaxation services offered by the Internet, while neurosis was negatively related to information services. In the case of women, the opposite happens, and extraversion obtained a negative score, and neurosis a positive score in terms of the use of social services. These results are important because they further confirm the connection between personality and behavioral analysis on the Internet. One of the authors of the study, Amichai-Hamburger, argued that the only way to outline the personality perspective more accurately on the use of Internet services would be a collaboration between IT people and psychologists. Given that the Internet is fueled by human interaction, it cannot be understood and developed if the personality of the target group is not known.

Moreover, in 2008, Amichai-Hamburger et al. [15] they researched the connection between personality and activity on nostalgic sites. Contrary to expectations, the results showed that extroverts were more active on nostalgic sites than introverts. The authors support the idea that online behavior also reflects real-life behavior. Therefore, extroverts maintain their dominance even in the virtual environment, compared to introverts.

In addition to the studies described above, Ross et al. [16] found a negative score for conscientiousness in terms of the number of activities on Facebook, as these people reduce the use of these networks to focus on deadlines and obligations. The results of this work were re-examined by Amichai-Hamburger et al. [17] and found that conscientious people generally have a larger number of friends, Using the Covariance Analysis (ANCOVA) to analyze the effect of conscientiousness on Facebook use, it was found that people who scored higher on the conscientiousness feature demonstrated a lower use of the image upload function on social networks.

In addition to these works, the Big Five model (Extraversion, Pleasure, Neurosis, Conscientiousness, Openness to Experience) was used as a reference by Mitja et al. [18] which demonstrated that a Facebook profile reflects a person′s real personality rather than their idealization. This personality model has been previously analyzed and discussed by Golbeck et al. [19] based on Twitter posts, considering the frequency of swearing and demographic information from the user′s profile. Non-linguistic features were also examined, such as the number of followers and the number of comments. For this analysis, LIWC was used—linguistic inquiry and word count—linguistic analysis and word counting, a database of the Medical Research Council (MRC) containing 150,000 words with linguistic and psycholinguistic features, but also two regression models: the Gaussian and ZeroR process.

In another paper, Verschuren [20] hired five psychology students to evaluate 65 LinkedIn profiles. These assessments were subsequently validated using a direct user study. Contrary to this manual approach, this paper provides an automated method for determining personality traits using data from Facebook and Twitter.

Additional research that needs to be mentioned is [21], which presents mapping between an ontological approach and social networks. However, this study is based on Hans Heysenck′s English and three-factor personality model (PEN) [22]: psychosis, extraversion and neurosis.

The approach of Andry Alamsyah et al. [23] is very interesting and very close to the methods covered in this paper. Alamsyah built an ontological model for the vocabulary of the Big Five personality model built in Bahasa, Indonesia.

Most of the above-mentioned research collected data from a single social network, whereas our research has the following main differentiators:Applying the personality analysis process on two social networks: Facebook and Twitter.Framing users in the DISC behavioral model.Using the Romanian language to create specific vocabulary in the form of an ontological model.

A comparison between these studies and the present paper is highlighted in Table 1. The goal of this table is to emphasize that linking personality types and activity on social networks is a topic of interest to researchers and different studies have been conducted. However, there is no similar study available for the Romanian language, as most of them focused on the English language for which more semantic developments are already available.

## 3. The Architectural Design of the Personality Type Segmentation System

The main objective of this research is to design and develop an application that identifies the behavioral profile of Social Media users by integrating semantic technologies for semantic analysis of statements and posts, based on the DISC model.

The application we propose in this paper is innovative in several aspects, such as:It is entirely designed for the Romanian language, where there have been little advancements regarding semantic processing and statement analysis.We propose and design an ontological approach for the data model of the personality types according to DISC.We propose the development of a Romanian vocabulary, which will be interconnected with the personality type data model, to illustrate the specific vocabulary and speech patterns used by each of the four personality types.

The development of the system involves four important steps (Figure 1):Develop an ontological data model for the DISC behavioral model. The ontology includes the specific behavioral patterns, needs, perceptions, specific vocabulary and communication baseline for each personality type.Develop a lexical database/vocabulary for the Romanian language and integrate it with the DISC ontological data model. The vocabulary will mainly focus on the specifics of the four personality types and will model a baseline vocabulary and speech pattern for each of them.Develop the statement analysis tool which collects social media posts from users (Facebook and Twitter), performs semantic analysis and classifies the user’s personality type according to DISC, based on statement analysis techniques.Provide a segmentation dashboard which can be used by recruitment and marketing specialists to target and position their campaigns more accurately. Based on the result, the application can make recommendations about what the user would like to see and how he/she should be approached.

### 3.1. The DISC Behavioral Model

In 1970, the American psychologist John G. Geier developed a model based on the behavioral research of healthy people, starting from the study of Moulton W. Marston—“Emotions of Normal people” [24]. Psychologist Geier described four fundamental types that are found in every human being, in different proportions. He argues that a consequence of human behavior would be the influence of two major directions of human perception of the environment and the relationship of individuals to it:Individuals perceive the environment as friendly or hostile.Individuals feel stronger or weaker in that environment.

This was the starting point for the development of the two-axis model, with the following poles: the perception of the environment as friendly or hostile and the perception of one’s own person as stronger or weaker in that environment. These poles have taken other forms over time, namely: the reaction to the environment—determined or restrained, and the perception of the environment—stressful or non-stressful. In the end, the DISC model identifies four personality types, as follows (Figure 2):

Table 2 summarizes the main characteristics of each personality type in the DISC model.

Currently, it has been proven that the DISC model is a very useful resource in companies, as the classification of the four types of personalities is based on two fundamental axes outlined around teamwork, tasks and duties: orientation towards people or towards the accomplishment of tasks and own fast or moderate rhythm [5]. Everyone has behavioral tendencies that belong to each individual behavioral style. However, based on the personal or professional environment, there is a tendency for a certain style to be more visible or used more frequently than the other three.

### 3.2. The Ontological Model for DISC

The first stage of the project consisted in the realization of the ontological model. To create the ontological model, we used Protégé [25], a free, open-source tool that can translate the structure of the ontology into the formal OWL language. The ontological model consists of:A general class that defines the concept of personality.Subclasses that define the personality types of the DISC model.The instances for each subclass, which are represented by the vocabulary specific to each personality type.

This model is depicted in Figure 3 below. The DISC ontology has four main classes, corresponding to the four personality types of the model (listed in the order from the figure—left to right):Dominant.Compliant.Steady.Influent.

Each of these classes has a set of instances associated (Figure 4), which define the specific vocabulary. The following figure shows examples of instances for the Dominant personality, but also the possibility of an instance belonging to several classes. For example, the word “involved” which could be specific to both a dominant and influential person.

For each class, the ontology contains the following number of instances, specific to the Romanian language: Dominant—146, Influential—115, Steady—101, Compliant—106. Table 3 provides some examples of instances used for each personality type (the ontology contains the Romanian words and phrases):

### 3.3. The System Architecture

Figure 5 depicts the overall architectural design of the personality type segmentation system based on the DISC model.

The system has three main components, as depicted in Figure 5:The Data Collection Engine, which collects input from social media posts (Facebook and Twitter). There will be a pool of social media users and the statement analysis will be conducted for each individual user.The Classification Engine—the core engine of the tool, which contains the ontological data model of the DISC personality, including the lexical ontology for the Romanian language, in order to provide a semantic analysis of the social media statements.

Once the data collection process is over and the input gets into the classification engine, the following steps will be necessary to obtain the personality type segmentation:Apply a Romanian stemming algorithm [4] which performs the following tasks: removes plurals, reduces the combining suffixes, removes standard suffixes, reduces verb suffixes.Use a POS Tagger for Romanian [5], based on hidden Markov model-based part-of-speech tagger for the Romanian language.

Process phrases and words based on the ontological model to extract semantic meaning and then to identify the personality type. A profile matching algorithm was designed and implemented in order to provide the statement analysis and personality segmentation.

3.The Data Visualization Engine, which provides the output as a dashboard containing two elements: the distribution, as a percentage, of personality types for each user and the user segmentation report, taking into consideration the highest percentage of a specific personality type for each user.

The workflow of the system is the following, depicted in Figure 6.

### 3.4. The System Development and Components

For the development of the application, we used a semantic data mining approach, based on ontological explicit specification of conceptualization of the DISC personality model. We made this choice considering that ontologies emphasize concepts, their role and properties and the relationships between them in order to create a logical and consistent knowledge base. The design of the ontological model is based on the mapping of words that reflect personality types to components of the model. These collections of words are verified by experts in the field and used in the form of a dictionary for measuring personality types. The ontological model thus created is a representation of the knowledge of an expert and facilitates the mapping of several traits in a complex sentence. The following setup was used, as depicted in Figure 7:

The core application was developed in Java [26] and it includes a data collection engine (further described in Section 3.4.1), a classification engine—which includes the data pre-processing module and the personality type classification module, which will be further described in detail in Section 3.4.2 and Section 3.4.3—and a Data Visualization Engine. The core application runs on an Apache Server [27] and is connected to the MySQL [28] database where the collected data are stored and to the ontological model, which is queried by the personality type classification module in order to establish the traits of a user.The MySQL database stores the user accounts and the datasets collected from the social media platforms. For the moment, it runs on localhost, on port 3306.The DISC ontological model is developed in Protégé [29]. We used the HermiT OWL Reasoner [30], an automated tool integrated in Protégé for the semantic validation of the ontology, which involves the automatic verification of the reasoning of the ontology, the detection of contradictions or logical inconsistencies. Based on an OWL ontology, HermiT can determine if the ontology is consistent, identifying the relationships between classes and instances. Figure 8 shows the semantic consistency and coherence of the ontological model that we developed, as confirmed by the HermiT Reasoner.
Figure 8Verification of the consistency and coherence of the ontological model.
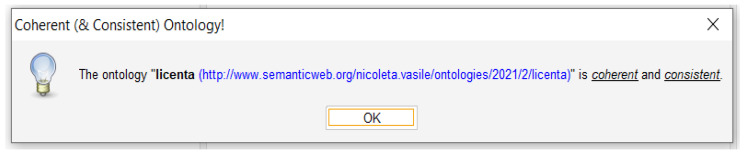


The core application developed in Java interacts with the ontological model created in Protege, through d the OWL API [31], which is used to create, manipulate and serialize ontologies written in OWL. The first step was to create an OWLOntologyManager instance [32], which handles the ontology management, as well as the mapping between the ontology and its correspondent OWL document. Afterwards, we created an OWLOntology object [33], and called the loadOntologyFromOntologyDocument method, to load the License.owl document, located in the project folder. Then we saved the instances of the classes in the ontology In a NodeSet <E extends OWLObject> structure, by calling the reasoner.getInstances method (part of the OWLReasoner). Using the getFlattened method, we took the instances from the nodes of the NodeSet structure and saved them in a structure of type Set <E extends OWLObject>.

#### 3.4.1. Data Collection from Social Networks

The application can collect posts from Facebook and Twitter.

A.Facebook

The permissions of Facebook Developer [34] are restrictive and only allow access to the posts of the person who created the application. In order to access other users′ pages, special permissions are needed, which explain in detail the application and the reason for using that permission. Thus, we only collected posts from the Facebook account of the authors of this paper. We used Selenium WebDriver [35] to simulate connecting to our Facebook accounts from a browser—in our case Google Chrome—and to automatically take over the access token to the data on Facebook.

From our Facebook accounts, we collected the following set of data (Table 4):

B.Twitter

Users mainly use Twitter to provide short descriptions of daily routines, for chatting, reporting news and exchanging information. Therefore, this platform provides a valuable opportunity to investigate personality expressions.

For Twitter, we were able to extract posts for different users with public accounts. To access the data on Twitter, we used Twitter4J [36], an open-source Java library that provides an API for accessing the Twitter API. Twitter4J provides the following types of actions:Posting a tweet.Retrieving a user′s timeline.Sending and receiving messages.Search for tweets.

#### 3.4.2. Data Pre-Processing

After the input data are collected from the social networks, they undergo a pre-processing phase, which is very necessary and important to delete irrelevant data and keep only meaningful information, in order to improve the quality of the data to be analyzed. Thus, certain barriers that might appear in the analysis process are reduced, and this stage also impacts the performance of the application, as the response time is improved.

The pre-processing steps are described in the following table (Table 5).

#### 3.4.3. Personality Type Classification

In order to identify the type of personality of a user, the words in the sample posts retrieved from Facebook and/or Twitter are analyzed and matched against the ontological model. Whenever a match is found between one word and an instance of the model, those posts are grouped into that personality type where the instance is found, and the counter is increased for that personality type. 

The workflow for the personality classification process is depicted in Figure 9.

For classifying the words according to the four types of personalities, we created a function that iterates through all instances of each personality type and, using the contains () method, we checked for the existence of the word among those instances, and we used a counter to save their number of occurrences. We obtained:D = number of words in the post assigned to Dominant.I = number of words in the post assigned to Influent.S = number of words in the post assigned to Steady.C = number of words in the post assigned to Compliant.

Then we added the number of words that fall into the Dominant type, with those under Influential, Steady and Compliant and we calculated a percentage for each of them, according to the following formulas:
Sum = D + I + S + CScore_Dominant = (D * 100)/sumScore_Influent = (I * 100)/sumScore_Steady = (S * 100)/sumScore_Compliant = (C * 100)/sum

The Algorithm 1 is the following:

**Step 1.** The user logs in to the application with a username and a password. If the user does not have an account, he can register by filling in the specific form.

**Step 2**. The user chooses the specific social media that will be used for analysis, namely Facebook or Twitter. If the user selects Facebook, then all posts will be retrieved from the account (as the API allows it). If the user selects Twitter, then the latest 100 posts will be retrieved (as the API is limited to this number for a free usage).

**Step 3.** The user will enter the username of the person he wants to analyze and then the connection to the social media will be made and the posts will be extracted. The sample shows the code used for retrieving posts from Twitter.
**Algorithm 1:**public void authAndGetTwitterPostsOnAction() throws TwitterException { String twitterUsername = getTwitterUsername.getText(); TwitterClass.authOnTwitter(); List<Status> statusesOfTwitterUser = TwitterClass.getTwitterPosts(twitterUsername);  for( Status status: statusesOfTwitterUser){  displayTweet.appendText(status.getText());  displayEditedTweet.appendText(TwitterClass.getEditedTweet(status.getText())); } tweetsWords = TwitterClass.getWordsFromTweets(statusesOfTwitterUser); System.out.println(tweetsWords); labelNoOfWordsFromTweets.setText("The number of words from tweets is: " + TwitterClass.getNumberOfWordsFromTweets(tweetsWords));}

At this step, the total number of the words from the posts is also calculated (Algorithm 2).

**Step 4.** The retrieved posts go through the pre-processing stage, as described in Section 3.4.2, Table 5.

**Step 5.** The results of the cleaned-up texts are then passed to the classification engine, where the following elements are determined:(a)The type of personality.(b)The number of words identified in the posts for every type of personality.(c)The score of every personality.

Step 5.1. At this step, the ontology created is loaded and the instances of every personality are extracted from the ontology model and the total number of them is calculated. The instances are stored in an ArrayList<String> variable. This can be observed in the code sample below.
**Algorithm 2:**public static ArrayList<String> getDominantIndividuals() throws OWLOntologyCreationException { OWLOntologyManager manager = OWLManager.createOWLOntologyManager(); // load ontology File inputOntologyFile = new File("Licenta.owl"); OWLOntology ontology = manager.loadOntologyFromOntologyDocument(inputOntologyFile); OWLDataFactory dataFactory = manager.getOWLDataFactory(); OWLReasonerFactory reasonerFactory = new ReasonerFactory(); OWLReasoner reasoner = reasonerFactory.createReasoner(ontology);reasoner.precomputeInferences(InferenceType.OBJECT_PROPERTY_ASSERTIONS,InferenceType.OBJECT_PROPERTY_HIERARCHY); OWLClass dominant = dataFactory.getOWLClass(IRI.create("http://www.semanticweb.org/nicoleta.vasile/ontologies/2021/2/licenta#Dominant")); //accessed on 1 September 2021 NodeSet<OWLNamedIndividual> individualsNodeSet = reasoner.getInstances(dominant, true); Set<OWLNamedIndividual> individuals = individualsNodeSet.getFlattened() for (OWLNamedIndividual ind: individuals) {  String individual = ind.toString();  individual = individual.substring(individual.indexOf("#") + 1, individual.length() −1);  if(individual.contains("_")){   String[] parts = individual.split("_");   for(int i = 0; i < parts.length; i++){    instancesDominant.add(parts[i]);   }  }else{   instancesDominant.add(individual);  } } return instancesDominant; }

Step 5.2. The words from the social media posts are compared against the words from the ontology model using a method that obtains as a parameter a variable of type ArrayList<String> which represents the words from the posts. With the help of a counter, we count the number of that similar words corresponding to each personality type (Algorithm 3).
**Algorithm 3:**public static int getNumberOfSimilarWords(ArrayList<String> wordsFromPosts) throws OWLOntologyCreationException { int counterDominant = 0; instancesDominant = getDominantIndividuals(); System.out.println("Similar words Dominant:"); for(String word: wordsFromPosts){  if(instancesDominant.contains(word)){   System.out.print( word + ",");   counterDominant++;  } }  System.out.println();  System.out.println("counter dominant:" + counterDominant);  return counterDominant; }}

Step 5.3. The score of each personality type is computed as the percentage of how many words were found in the posts for every personality type, as shown in the code sample below for calculating the percentage score for each personality type (Algorithm 4).
**Algorithm 4:**public float getScoresOfDominant() throws OWLOntologyCreationException {  float scoreDominant = 0;  float dominanats = Dominant.getNumberOfSimilarWords(tweetsWords);  float sum = getSumOfSimilarWords();  if (getSumOfSimilarWords() == 0) {   System.out.println(“Sum is 0”);  } else {     scoreDominant = (dominants * 100)/sum;    }  return scoreDominant; }

The results will be shown as a table, as shown below (Table 6):

These results show that an individual tends to have a dominant personality type, but also has characteristics of other personality types. The outcome of the process is to obtain a dashboard with users’ personality type, which will be used to plan and implement recruitment and marketing campaigns, to correctly and accurately target and position interactions customized to each type of candidate or customer.

## 4. Results and Analysis. Discussion

### 4.1. Results Generated by the Application

#### 4.1.1. The Datasets Used

For the validation of the model developed we used a dataset of almost 2000 posts retrieved from 10 social medial accounts (Facebook and Twitter). The dataset is in text format, entirely in Romanian language. The profile of the users is diversified, gender balanced (five female, five male), as we selected public social media profiles with various backgrounds in order to validate the results on general use cases. From Facebook, we used the profiles of the co-authors of the papers, while for Twitter, we selected seven profiles of public figures coming from media, entertainment and journalism.

The two datasets have the following specifications:A.Facebook

We used a set of 998 posts collected from the accounts of the three co-authors of this paper, all written in Romanian. A total number of 19,950 words were analyzed for this dataset. The sample source code used to retrieve posts from the Facebook accounts is the following (Algorithm 5):
**Algorithm 5:**FacebookClient fbClient = new DefaultFacebookClient(accessToken, Version.VERSION_9_0); User user = fbClient.fetchObject("me", User.class); username = user.getName(); Connection<Post> result = fbClient.fetchConnection("me/feed", Post.class); int counter = 0; for (ConnectionIterator<Post> iterator = result.iterator(); iterator.hasNext();) {  List<Post> page = iterator.next();  for (int i = 0, pageSize = page.size(); i < pageSize; i++){   Post aPost = page.get(i);   myPosts.append(aPost.getMessage());   counter++;  } }

B.Twitter

We used seven samples (700 tweets, ~35,000 words) collected from public users of the social network Twitter that meet certain conditions, namely:The user must be a public and well-known person in Romania and have a public account, so that we can compare the result of the analysis to the image created through online presence, media appearances or public events.The user must be active on Twitter and post consistently.The user must post primarily in Romanian.Selected to be from different fields of activity.

The sample source code used to retrieve posts from the Twitter accounts is the following (Algorithm 6):
**Algorithm 6:**public static List<Status> getTwitterPosts(String username) throws TwitterException { Paging paging = new Paging(1, 100); List<Status> statusesOfTwitterUser = twitter.getUserTimeline(username, paging); System.out.println(statusesOfTwitterUser.size() + "tweet-uri extrase"); return statusesOfTwitterUser;}

The following table (Table 7) depicts the users we have selected who meet the criteria, the number of tweets collected and the number of words in those tweets.

#### 4.1.2. The Results and Discussion

For the samples presented above, the application generated the following results (Table 8 and Table 9):A.Facebook dataset

These results will be further analyzed in Section 4.2.

B.Twitter dataset

For the samples collected from Twitter, we obtained the following results using our application:

The user @user1 has a high score on the Dominant feature, which means that he is a person oriented towards success, results and objectives. He is passionate about change and variety and loves competition and risk. Among the words in the tweets collected from his page that fall on the Dominant side are the following: support, propose, start, power, intelligence, endurance, money, careers, responsible, base, communicate. Based on his public image, we consider this result to be close to reality, as he is a man passionate about technology and change, involved in many projects and eager for results.

The user @user2 has a high score on the Influent side, which characterizes him as an open, cheerful person, eager for the public and attention, teamwork, who easily externalizes and appreciates the chance to freely express ideas and feelings. Among the specific words for Influent identified in his tweets are the following: humor, good, together, joy, we are, speak, attention, pleasure, vote. Based on his public image and media and online appearances, he is an influential person, close to his public and who encourages free expression of thoughts and emotions.

The user @user6 has a high score on the Compliant side, which describes her as a calm, calculated person, someone who sets high standards, but who is sentimental and emotional and prefers meditation. Among the words that framed her in this area of personality are the following: thank you, heart, evening, relaxed, pleasant, lonely, art, beautiful, we love, thought, hug, loved ones, people, soul. Based on her public image, the evaluation is realistic, judging by the fact that she is a romantic artist, and her musical performances are full of emotion and feelings.

The user @user4 is also part of the Steady personality profiles, which describe him as a man who likes stability and security, who is open to new ideas, likes collaborations, exchanging ideas and experiences. The following words specific to the Steady profile were identified in his tweets: relationships, grateful, inspires, together, feel, world, happy, support, constant. Based on his public image, we evaluate the report to be realistic.

### 4.2. The Validation of the Model

The validation process is essential to prevent and minimize doubts about the classification process. We turned to psychologists to validate the classification results, for them to ensure that each classified word belongs to the correct personality, because Romanian words often have different meanings depending on their contextual purposes.

#### 4.2.1. Comparison with Certified DISC Assessments

The first step in assessing the validity obtained with our platform was to compare the results generated with a set of results obtained by taking certified DISC assessments, validated by psychologists. We compared the results generated by the platform for the three co-authors of this paper with the results obtained from two online assessments for the DISC profile [39,40].

This result suggests that @author1 has specific traits of the two personality types (Figure 10), Steady and Compliant, which supports the result generated by our application. The words that best describe her would be useful, pleasant, logical and methodical.

This result suggests two dominant personalities, namely Steady and Compliant (Figure 11), which also supports the classification provided by our application.

The results obtained for @author2 also validate the evaluation generated by our platform in terms of DISC profiles. As we can notice in Figure 12, her dominant personality type driving her behavior is Dominant, just as our application determined. 

Table 10 provides a comparative overview of the results generated by our platform and the results obtained when taking certified DISC assessments for the three co-authors. We consider this to be a strong validation for our work, as the certified tests that we took are approved by psychologists, thus their results can be considered as baseline for the personality traits classification.

#### 4.2.2. F-Score

The next step to assess the quality and robustness of the classification process consisted of using the FScore measure [41], which is a reference measure used in information retrieval and data classification to determine accuracy. It is calculated from the precision and recall of the test, where the precision is the number of true positive results divided by the number of all positive results, including those not identified correctly, and the recall is the number of true positive results divided by the number of all samples that should have been identified as positive. The formula used is the following:
(1)$$F-Score=2*\frac{precision*recall}{precision+recall}$$

Roughly, precision answers the question: “How many of the elements in this cluster belong there?”, whereas the recall answers the question: “Did all of the elements that belong in this cluster make it in?”. A perfect clustering solution will be the one in which every class has a corresponding cluster containing exactly the same documents in the resulting clustering. In this case, the FScore will be one. The higher the FScore value, the better the clustering solution is.

Figure 13 presents the workflow of the test method and validation process based in the F-Score measure.

Table 11 shows the F-Score values for the tests performed on the dataset collected for the 10 users selected—2000 Facebook and Twitter posts in total, in the Romanian language. For each user, we computed an F-Score based on the classification obtained using our platform for each of the four personality types.

The platform produced good results during all tests which were conducted in this work. It proved that it can correctly identify the personality traits of a user based on the semantic analysis of their social media posts in Romanian, without any prior information. Therefore, the ontological model that we developed has proven its capability to evaluate the personality classification and thus foster cyber physical social systems.

## 5. Conclusions

Social networks have become more and more of a way to share our daily lives. Digital footprints, posts, photos and comments can expose a person′s personality, and the analysis of users’ behavior can play a significant role in establishing the social space in the context of the emerging cyber-physical social systems. 

In this research, we explored a semantic approach to analyze the behavior of users on the social networks Facebook and Twitter, to build an ontological model for measuring human personality, namely the classification of individuals according to the personality types of the DISC model (Dominant, Influential, Stable, Conscious). The vocabulary is built entirely for the Romanian language and the model underlying the classification of individuals is built on two axes, namely: perception of the work environment and how people to relate to it. Thus, this type of application can provide support in recruitment processes on the side of psychological interviews or can help the employer to much more quickly and easily shape the profile of a potential employee, to establish compatibility between the individual and the workplace and to integrate new employees in the organizational culture. 

As future developments, we envisage the following:Adding the necessary permissions to be able to extract posts from the pages of any Facebook user.Enriching the vocabulary of the ontological model to improve the accuracy of the analysis and classification.Adding Instagram and Twitter to the list of social networks from which data are retrieved.Improve accuracy by analyzing conversational data and posts over a longer period, to ensure that users are constantly using similar words to show their feelings or interact with others.Boost dynamic collaboration in the context of CPS through customer engagement via social media and tools, based on personality measurements.

## Figures and Tables

**Figure 1 sensors-21-06611-f001:**

The 4 development steps for the application.

**Figure 2 sensors-21-06611-f002:**
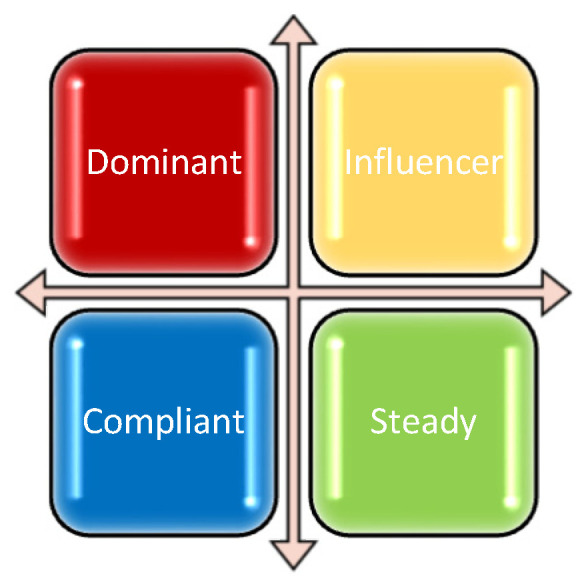
The 4 behavioral types of the DISC model [4].

**Figure 3 sensors-21-06611-f003:**
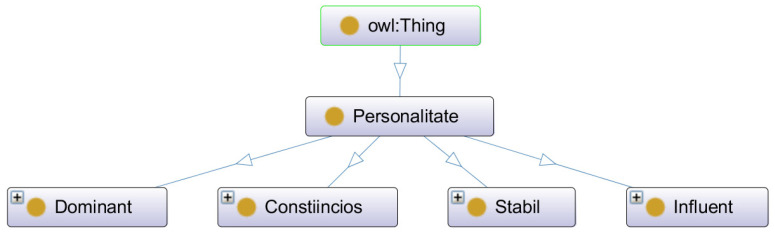
The ontological model for DISC.

**Figure 4 sensors-21-06611-f004:**
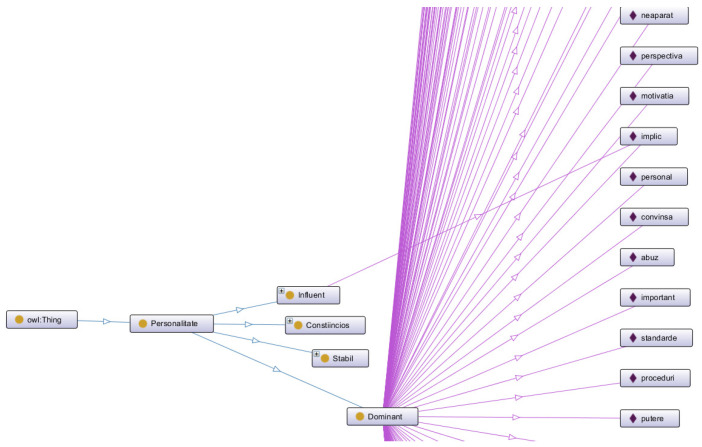
Example of instances of ontological model classes.

**Figure 5 sensors-21-06611-f005:**
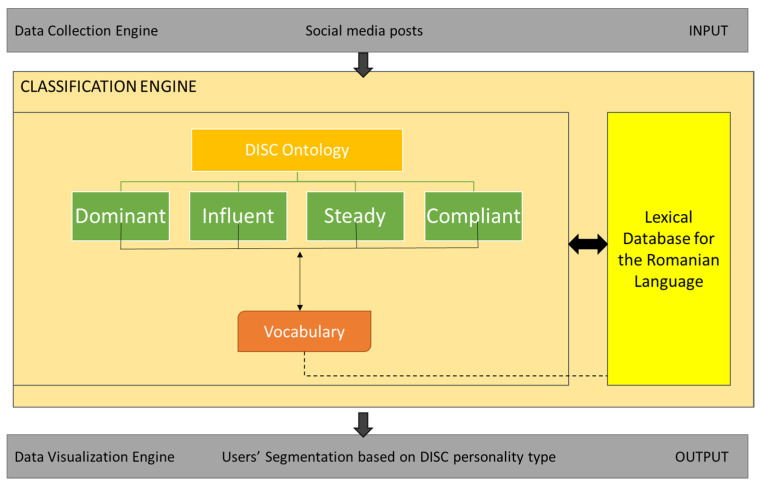
Architectural design of the personality type segmentation system.

**Figure 6 sensors-21-06611-f006:**
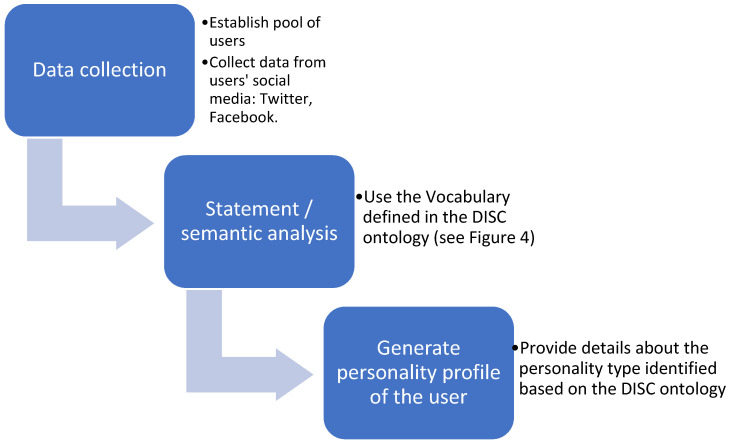
The flow of the application.

**Figure 7 sensors-21-06611-f007:**
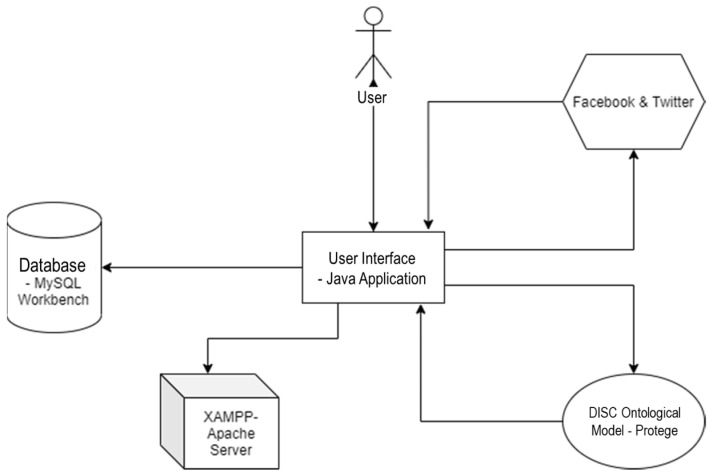
The development setup and components diagram.

**Figure 9 sensors-21-06611-f009:**
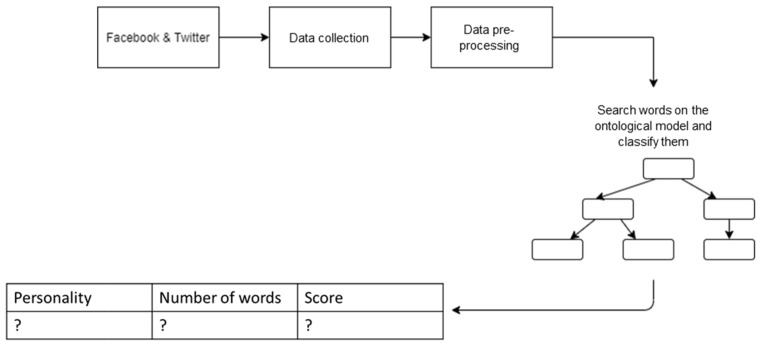
The flow for personality classification.

**Figure 10 sensors-21-06611-f010:**
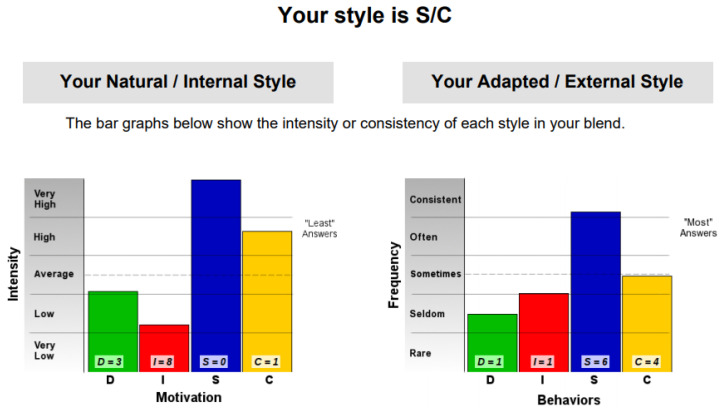
The DISC classification provided by [39].

**Figure 11 sensors-21-06611-f011:**
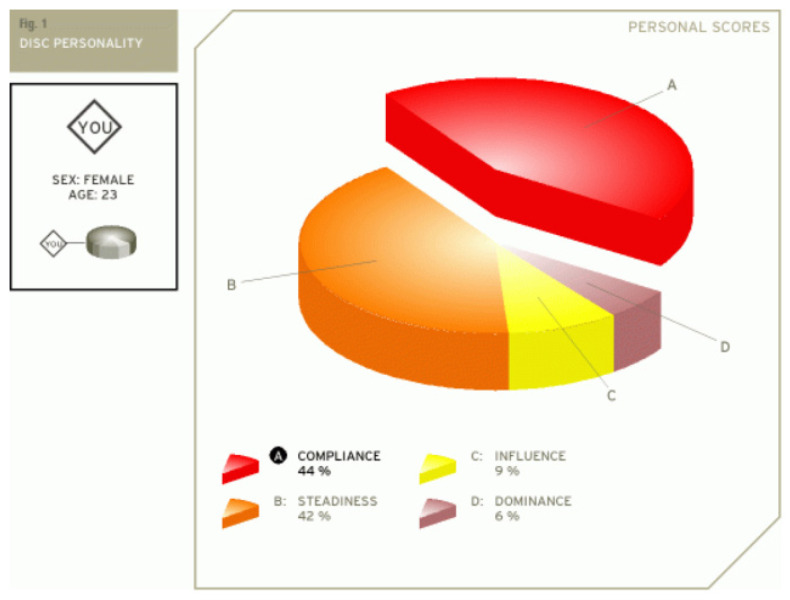
The DISC classification provided by [40].

**Figure 12 sensors-21-06611-f012:**
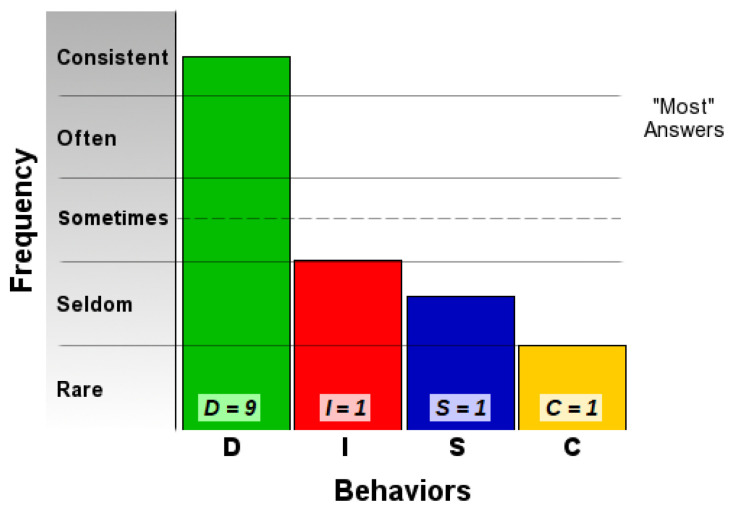
The DISC classification provided by [39].

**Figure 13 sensors-21-06611-f013:**
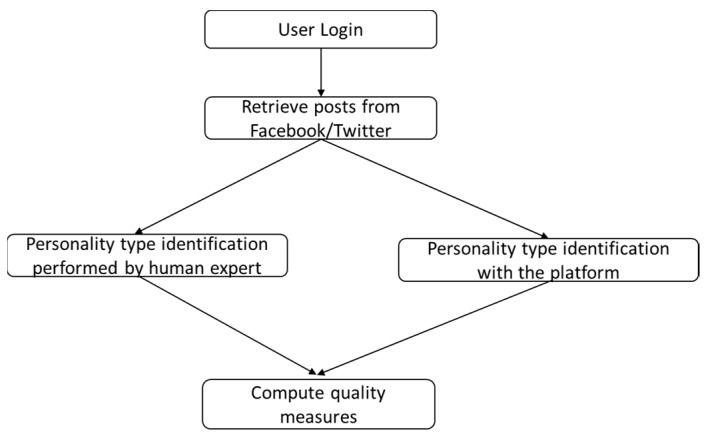
The test method and validation flow.

**Table 1 sensors-21-06611-t001:** Comparison between various studies and the present research.

Study	Examined Personality Traits/Personality Patterns	Social Networks	Approach	Language	Comments
Amichai-Hamburger [14]	Extraversion and Neurosis	Internet services for relaxation, socializing or information	Manual study	English	Demonstrated the connection between human personality and the use of the Internet
Amichai-Hamburger et al. [15]	Extraversion and Introversion	HEVREA – nostalgic site	Manual	English	The dominance of extroverts that is reflected both in real life and in the online environment, on sites prone to introspection
Rose et al. [16]	Big Five Model	Facebook	Manual study	English	It has been found that extroverts are part of several groups on Faceboook, and neurotics have more posts, fewer photos and tags
Amichai-Hamburger and Vinitzky [17]	Big Five Model	Facebook	Automated, Covariance Analysis (ANCOVA)	English	Conscientious people have a large number of friends on Facebook, but activity on this network is low.
Golbeck et al. [19]	Big Five Model	Twitter	Automated, LIWC—linquistic inquiry and word count, 2 regression algorithms: Gaussian process and ZeroR	English	Building a model that can predict the percentage for each personality type in the Big Five model with an accuracy between 11–18%
Verschuren [20]	Big Five Model	LinkedIn	Manual (with the help of assessments made by psychologists)	English	The complexity and completeness of the LinkedIn profile influences the accuracy of the results in terms of the type of personality of the user
Andry Alamsyah et al. [23]	Big Five Model	Twitter	Automated, Ontological Model	Bahasa, Indonesia	It highlights the existence of several dominant personality types in terms of an individual
The present study	DISC	Facebook and Twitter	Automated, Ontological Model	Romanian	Analysis of posts on social networks and calculation of the percentage for each personality type in the DISC model

**Table 2 sensors-21-06611-t002:** The main characteristics of DISC personality types in the DISC model [4].

Behavioral Style	Main Characteristic	Description
Dominant	Dominance	They are extroverted and goal oriented. People with this type of behavior appreciate challenges and struggle to overcome any obstacles. They are motivated by power and control and they like competition. Decisive reaction to the environment, which is perceived as hostile. They accept challenges and want to win. As communication style, they are very direct and sharp and often do not show empathy for those around them.
Influencer	Inducement	They are extroverted and oriented towards relationships with others. They achieve their goals through relationships and forming alliances. Determined reaction to the environment, which is perceived as friendly. They prefer to work in a team and try to convince and influence others, rather than impose things on them. Need to motivate others, to express themselves and to be heard.
Steady	Submission	They are introverted and relationship oriented. They are motivated by safety, appreciate a clear working style, based on well-defined rules and processes. They are good planners and strictly follow established procedures. They do not like changes and do not show initiative, preferring the execution area. Retained reaction to the environment, which is perceived as friendly. Need for stability and harmony. They want to support others and build stable relationships.
Compliant	Compliance	They are introverted and task oriented. They prefer order and discipline. He focuses a lot on details, thinks very critically and works well in very clearly defined conditions, with established norms and standards. Retained reaction to the environment, which is perceived as hostile. Need to do everything right. He wants to avoid problems with accuracy and precision.

**Table 3 sensors-21-06611-t003:** Examples of instances for each class of the DISC ontological model.

Personality Type	Instances
Dominant	now, necessarily, results, goals, immediately, must, benefit, courage, determined, impatience
Influent	Together, lovely, jovial, people, story, cheerful, say, talk, interest, inspiration, discussion
Steady	Appreciation, ambition, compassion, consistent, responsibility, solidarity, simplicity, meeting, grateful
Compliant	I analyze, research, meditation, listen, time, clear, dear, beautiful, thought, peace, organization, how long does it take

**Table 4 sensors-21-06611-t004:** Users and data collected from Facebook.

Facebook User	Number of Posts	Number of Words to Be Analyzed
@author1	321	6420
@author2	530	10600
@author3	147	2205

**Table 5 sensors-21-06611-t005:** Pre-processing steps.

Step	Procedure	Description
1.	Lowercasing	Turns all capital letters into lowercase letters. This is standard practice in text mining, it is efficient in dealing with sparsity issues and helps increase the consistency of the outcome. The toLowerCase () method was applied to a string containing the posts, and it returned another string, but no capital letters.
2.	Remove URLs	Removes all URLs from the text. The Java Matcher class (java.util.regex.Matcher) was used to search the text for occurrences of a regular expression and replace it with “ ”.
3.	Remove diacritics	Diacritics, special characters specific to the Romanian language, were removed from the text to obtain generalized data, using the stripAccents () method in the StringUtils class. This step helps in removing noise before the classification process.
4.	Remove special characters	We removed any non-letter special characters from the text. We used the replaceAll () method, with which one regular expression is replaced by another. The regular expression is “[^ a-zA-z]”. This step helps in removing noise before the classification process.
5.	Remove null	We noticed that after collecting the posts from social networks, the text contained the null string which probably replaced a character or symbol that could not be interpreted and displayed. We chose to remove these strings to improve the performance of the analysis using the replaceAll () method. This step helps in removing noise before the classification process.
6.	Stemming algorithm	Stemming is a text pre-processing technique to reduce variation in words to their root form. This step enhances the standardization of the vocabulary which will be further used in the classification process. We applied a Romanian stemming algorithm [37] which performs the following tasks: removes plurals, reduces the combining suffixes, removes standard suffixes, reduces verb suffixes.
7.	Part of Speech (POS) tagger	POS tagging is the process of mapping a word in a text to its corresponding part of speech, by also considering the context. We used a POS Tagger for Romanian [38], based on hidden Markov model-based part-of-speech tagger for the Romanian language. This component assigns parts of speech for each word in the text, such as noun, adjective, verb and so on. This step leverages word sense disambiguation, as the POS tagger can help differentiate between the two meanings of the word. Previous tests [38] demonstrated a 90–95% accuracy of the POS tagger.

**Table 6 sensors-21-06611-t006:** Sample of results generated for a user.

Personality	Number of Instances from the Vocabulary	Number of Words from Posts	Score
Dominant	146	42	22.340425
Influent	115	36	19.148935
Steady	101	84	44.688085
Compliant	106	26	13.829787

**Table 7 sensors-21-06611-t007:** Sample dataset from Twitter.

Twitter User	Number of Tweets	Number of Words to Be Analyzed
@user1	100	1572
@user2	100	990
@user3	100	2661
@user4	100	1444
@user5	100	1420
@user6	100	1778
@user7	100	2808

**Table 8 sensors-21-06611-t008:** Results for Facebook dataset.

@User1	@User2	@User3
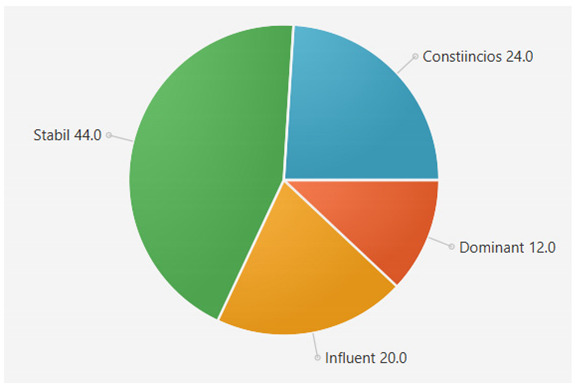	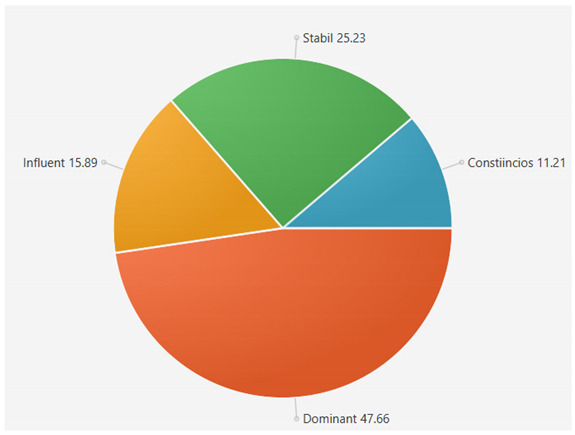	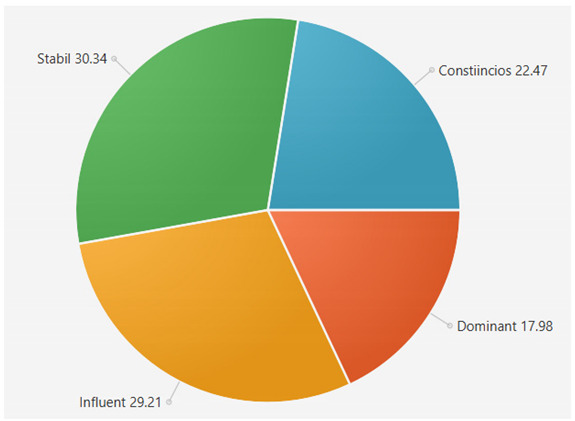

**Table 9 sensors-21-06611-t009:** Results for Twitter dataset.

Twitter User	Results
@user1	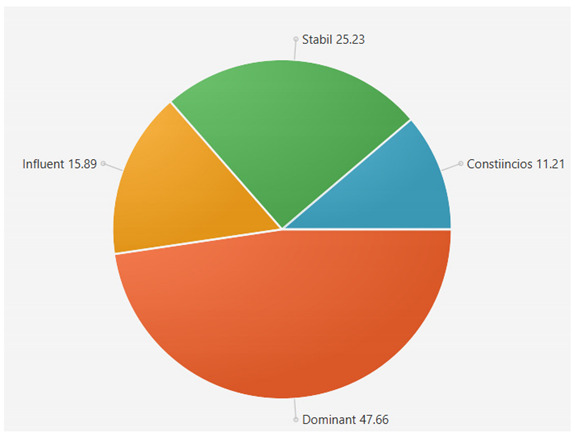
@user2	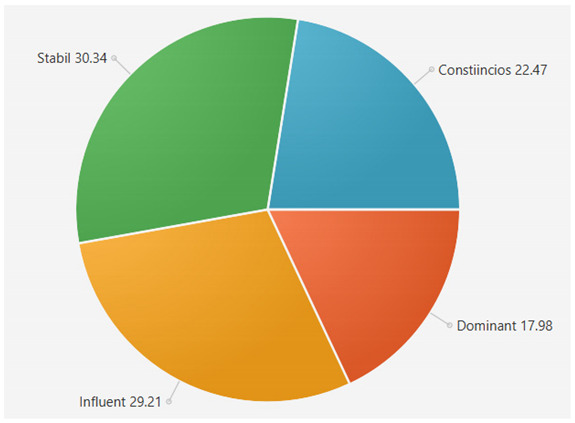
@user3	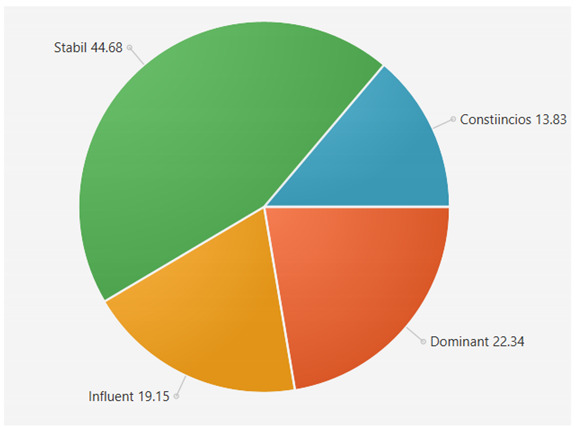
@user4	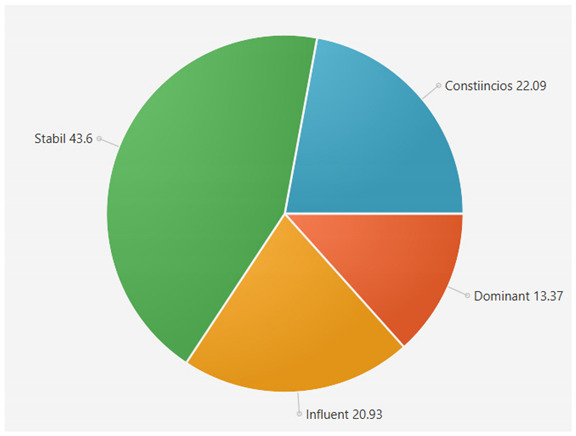
@user5	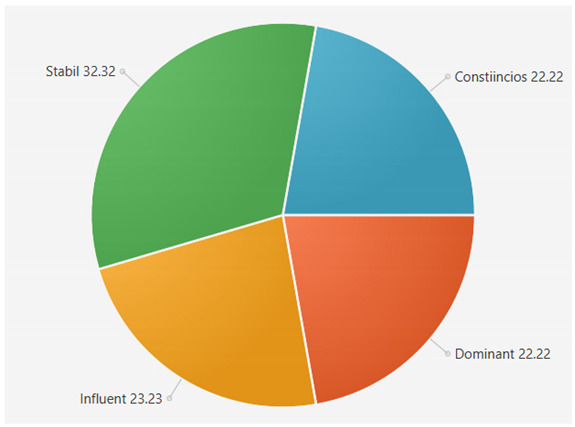
@user6	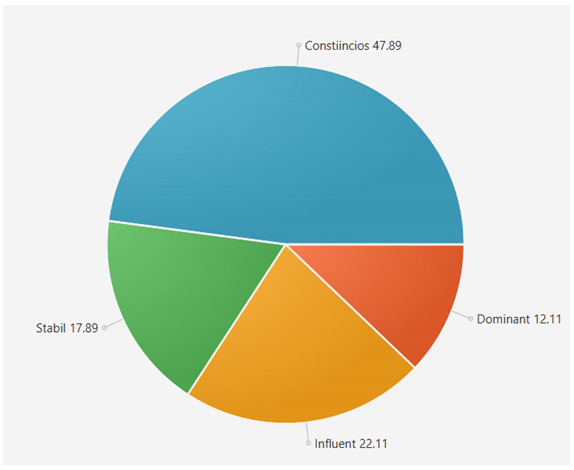
@user7	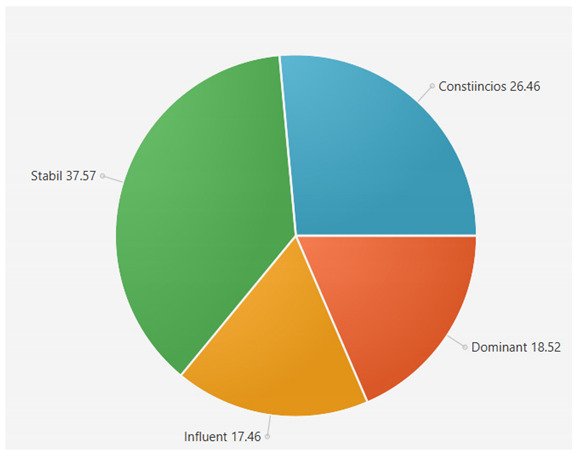

**Table 10 sensors-21-06611-t010:** Comparison of results with certified DISC assessments.

	Assessment 1	Assessment 2	Our Platform
@author1	Compliant 46%	Compliant 44%	Compliant 36%
	Steady 43%	Steady 42%	Steady 44%
	Influent 6%	Influent 8%	Influent 10%
	Compliant 5%	Compliant 6%	Compliant 10%
@author2	Dominant 48%	Dominant 45%	Dominant 48%
	Influent 17%	Influent 20%	Influent 15%
	Steady 17%	Steady 17%	Steady 20%
	Compliant 17%	Compliant 18%	Compliant 17%
@author3	Steady 32%	Steady 30%	Steady 30%
	Influent 33%	Influent 30%	Influent 30%
	Dominant 15%	Dominant 22%	Dominant 20%
	Compliant 20%	Compliant 18%	Compliant 20%

**Table 11 sensors-21-06611-t011:** Comparison of results with certified DISC assessments.

F-Score	Dominant	Influent	Steady	Compliant
@user1	0.89	0.92	0.95	1
@user2	1	0.96	0.95	0.92
@user3	0.98	0.97	0.98	0.98
@user4	0.87	0.82	0.91	0.93
@user5	0.90	1	0.96	0.93
@user6	0.83	0.91	0.88	0.86
@user7	0.92	0.97	0.88	0.94
@user8	0.87	0.93	0.98	0.98
@user9	0.94	0.95	1	0.98
@user10	0.90	0.93	0.88	0.92

## Data Availability

Not applicable.

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
