# Peer review of "Fostering Cyber-Physical Social Systems through an Ontological Approach to Personality Classification Based on Social Media Posts"

_sensors, 2021, doi:10.3390/s21196611_

Round 1

Reviewer 1 Report

The paper is interesting and addresses a relevant topic. The study is sound and the conclusions seem to be coherent. However, before the acceptance, I suggest some changes regarding the methods and materials description, because I think some information is missing.

First, the selected datasets must be explained much more carfully. Data formats, population, gender perspective, etc. must be discussed. It is essential to learn how general your results are. Besides, validity threats must be analyzed, because in this context are essential.

Algoritms must be clearly described using pseudo-code or a much more detailed chart. Now some mathematical expressions and qualitative descriptions are provided, but that's not enough.

The employed hardware and software tools have to be described. What kind of data mining tools did you employed? why? what was your hardware platform? some errors or bias could be caused by these materials?

Some statistical test need to be conducted. Anova, T-Students, etc. You need to compare your results an state of the art mechanisms, before arguing your solution is better. Now, you are only comparing some indicators, but that not solid enough. 

Author Response

We really appreciate all insightful comments and useful suggestions, which definitely will help us improve the quality and readability of our manuscript. All the suggestions and criticism are fully appreciated. All the comments are highly constructive and useful to restructure our manuscript. The manuscript has been revised according to the comments. We hope all the modifications will fulfill the requirement to make the manuscript acceptable for publication in this archive journal.

Reviewer 2 Report

The paper presents a personal classification tool based on the DISC personality types. Authors analyze data collected from posts on FB and Twitter. Authors introduce  the architectural design, the system architecture and how data are collected and the experimental results.

The paper is well written and structured, however some major issues prevent accepting the paper in the current form:

1) the state-of-art section reports a comparative table in which authors list relevant works. Most of the selected works are based on the English language, while the target language of authors is the Romanian language. Authors are encouraged to clarify if the target language (Romanian) also affects the design of the ontology and the system architecture. If this is the case, then authors have to clearly describe why the selected language modifies the ontology and the design, otherwise authors have to fairly compare with other works not only on the target language, rather on the design of the ontology and of the system architecture and on the followed methodology.

2) The proposed architecture and the model is reasonable, however they lack of requirements. It is not clear how to extract the personal traits reported in table 3, authors state “Use a POS Tagger for Romanian [5], based on hidden Markov model-based part-of-speech tagger for the Romanian language”  but more information are needed as this might be a crucial aspect

3) the personal traits classification reported in Section 3.4.3 seems very straightforward. The results do not provide evidence of its effectiveness in terms of accuracy, precision, recall and other metrics commonly used for the classification purpose.

4) The data sets used are not representative for the Romanian language. Authors extract a limited number of posts from personal FB and Twitter accounts. The limited number of data does not allow to stress the POS Tagger nor the methodology proposed in Section 3.4.3. Furthermore  it is not clear how the obtained results fro the trait classification is compared against a ground truth.

Author Response

We really appreciate all insightful comments and useful suggestions, which definitely will help us improve the quality and readability of our manuscript. All the suggestions and criticism are fully appreciated. All the comments are highly constructive and useful to restructure our manuscript. The manuscript has been revised according to the comments. We hope all the modifications will fulfill the requirement to make the manuscript acceptable for publication in this journal.

Round 2

Reviewer 1 Report

In general, all my previous concerns have been addressed and the paper may be accepted. 

Some minor changes are still pending, such (for example) replacing the screenshots by real lists with pseudocode.

Author Response

(The authors gave the same response as above.)

Reviewer 2 Report

Authors improved their paper. However, some major issues still remain valid:

1) authors have to specifically address comment 4 :

"The data sets used are not representative for the Romanian language. Authors extract a limited number of posts from personal FB and Twitter accounts. The limited number of data does not allow to stress the POS Tagger nor the methodology proposed in Section 3.4.3. Furthermore it is not clear how the obtained results fro the trait classification is compared
against a ground truth."

The answer provided by authors to comment 4 refers to comment 3, namely the metrics used for the performance evaluation. The limit dimension of the data set still is an issue. Authors have to deal with it and clarify is data set provide significant results or not. In the negative case, authors have to consider how to retrieve more data, by also following the GDPR regulation for the retrieved sensitive information

2) Authors reply similarly to different comments: ex: answers to comments 2 and 3 partially match, however comment 2 refers to requirements leading to such design, while comment 3 refers to how to measure the performance. In both of the comments, it was not asked to detail the steps of the implemented algorithms (comment for the other reviewer)

3) the screenshots of the code snippets (fig 10 11 12 13) are not recommended as they introduce a low-quality presentation. Authors can consider using pseudocode or code listings.

Author Response

(The authors gave the same response as above.)
